# Effectiveness of Physical and Therapy Interventions for Non-ICU Hospitalized Pneumonia Patients: A Systematic Review of Randomized Controlled Trials

**DOI:** 10.3390/healthcare13121444

**Published:** 2025-06-16

**Authors:** Takako Tanaka, Yorihide Yanagita, Tatsuya Morishita, Fumiya Nagata, María Granados Santiago, Laura López-López, Marie Carmen Valenza

**Affiliations:** 1Department of Physical Therapy Science, Nagasaki University Graduate School of Biomedical Sciences, Nagasaki 852-8523, Japan; tanakataka@nagasaki-u.ac.jp (T.T.); y-yanagita@nagasaki-u.ac.jp (Y.Y.); da1110kj@gmail.com (T.M.); 13rp22@g.seirei.ac.jp (F.N.); 2Department of Rehabilitation, Tagami Hospital, Nagasaki 851-0251, Japan; 3Department of Nursing, University of Granada, 18071 Granada, Spain; 4Department of Physical Therapy, University of Granada, 18071 Granada, Spain; lauralopez@ugr.es (L.L.-L.); cvalenza@ugr.es (M.C.V.)

**Keywords:** pneumonia, hospitalization, physical therapy, length of hospital stay

## Abstract

**Background:** Pneumonia is a leading cause of hospitalization worldwide, particularly among older adults and individuals with comorbidities. Physical therapy is increasingly recognized as a key component in improving health-related outcomes. The aim of this systematic review and meta-analysis was to evaluate the effectiveness of physiotherapy in patients with non-complicated hospital pneumonia. **Methods**: A systematic review and meta-analysis were conducted to determine the impact of physical therapy on clinical outcomes in non-ICU (Intensive Care Unit) pneumonia patients compared with usual care. The study followed the Preferred Reporting Items for Systematic Reviews and Meta-Analyses (PRISMA) guidelines and was registered in PROSPERO (CRD42024565419). PubMed, Scopus, Web of Science, and Cochrane Library were systematically searched to identify randomized controlled trials from their inception to July 2024. The Cochrane Risk of Bias Tool version 2.0 (RoB 2) and the Downs and Black checklist were used for quality assessment. A quantitative synthesis was performed using the Review Manager (RevMan) version 5 software program. **Results**: We selected nine studies, which included 1349 hospitalized non-ICU pneumonia patients. The results showed significant differences in favor of physical therapy interventions compared to usual care, with a reduction in length of hospital stay (*p* = 0.009). Also, there were significant differences in dyspnea levels (*p* < 0.00001). **Conclusions**: Physical therapy interventions may contribute to a reduced length of hospital stay and improved dyspnea levels in hospitalized non-ICU pneumonia patients.

## 1. Introduction

Pneumonia remains one of the most significant respiratory infections worldwide, contributing substantially to morbidity and mortality, particularly among vulnerable populations such as the elderly, immunocompromised individuals, and those with chronic comorbidities [1]. It is a leading cause of hospitalization, with severe cases often requiring intensive care and prolonged hospital stays [2]. The clinical course of pneumonia is frequently complicated by physical deconditioning, respiratory dysfunction, and reduced quality of life, both during the acute phase and the recovery period [3,4].

These challenges, combined with the loss of functional independence in patients surviving to hospital discharge [5], underscore the importance of comprehensive management strategies that address not only the infection itself but also the functional and psychological consequences of the disease and its treatment [6].

Hospitalization due to pneumonia frequently results in substantial physical deconditioning, primarily caused by extended periods of bed rest, diminished mobility, and the systemic inflammatory response triggered by the infection [7]. This decline in physical conditioning can further compromise respiratory muscle strength, worsen pulmonary function impairments, and prolong recovery time. Consequently, patients face an elevated risk of adverse outcomes, including hospital-acquired infections, unplanned readmissions, and persistent functional disability, all of which contribute to increased morbidity and long-term health burdens [8,9].

In recent years, there has been growing recognition of the role of physiotherapy, including respiratory and physical rehabilitation, has been shown to improve respiratory function, enhance physical capacity, and reduce the risk of complications during hospitalization and post-discharge [10,11]. Respiratory physiotherapy techniques, such as breathing exercises, airway clearance, and inspiratory muscle training, can help optimize lung function, reduce dyspnea, and prevent respiratory complications [12,13]. Meanwhile, physical rehabilitation programs, including early mobilization and tailored exercise interventions, can counteract the effects of deconditioning, improve muscle strength, and enhance overall functional capacity [14,15].

Integrating physiotherapy into the treatment plan for patients with pneumonia has been shown to accelerate recovery and reduce the impact of disability following hospitalization [16,17]. As a supportive intervention, physiotherapy contributes to this process by improving pulmonary function, enhancing airway clearance, and preventing complications such as atelectasis and deconditioning [18,19]. In addition, it plays a key role in encouraging patients to actively participate in their recovery, thereby strengthening self-efficacy and adherence to rehabilitation programs [20,21].

Current evidence supports the safety and effectiveness of physiotherapy in the management of critically ill patients with pneumonia, although there are limitations when considering the patient’s baseline physical and respiratory function, comorbidities, and personal goals [22]. Chen, X. et al. [23] observed the effect of chest physiotherapy on shorten hospital stays, fever duration, and Intensive Care Unit (ICU) stays, as well as mechanical ventilation. However, there is still a need for more robust evidence on the most effective approaches to delivering these interventions for patients admitted to hospital with uncomplicated pneumonia.

To date, no systematic review has comprehensively synthesized evidence on the role of physiotherapy in the management of pneumonia, particularly in the context of non-ICU hospitalization. For this reason, this systematic review and meta-analysis aimed to evaluate the effectiveness of physiotherapy in treating uncomplicated hospital-acquired pneumonia.

## 2. Materials and Methods

### 2.1. Study Registration

This systematic review adheres to the guidelines outlined in the Cochrane Handbook for Systematic Reviews [24] and follows the checklist provided by the Preferred Reporting Items for Systematic Reviews and Meta-Analyses (PRISMA) statement [25]. The protocol for this study was registered under the International Prospective Register of Systematic Reviews (PROSPERO) with the number CRD42024565419.

### 2.2. Search Strategy

We systematically searched the PubMed, Scopus, Web of Science and Cochrane Library databases for published studies from inception to July 2024. The MEDLINE search strategy was systematically constructed via a three-phase process: (1) keyword derivation through analysis of existing systematic reviews, (2) MeSH term optimization via database interrogation, and (3) expert-led refinement. To ensure sensitivity, reference lists of key reviews were hand-searched, and the strategy was iteratively tested and adjusted for cross-database compatibility.

Following this, the strategy was adjusted for indexing across each database. A comprehensive description of the search strategy is provided in the Appendix A.

A broad search strategy was employed to maximize the retrieval of all potentially relevant studies, acknowledging that this approach may result in a high initial number of records but helps minimize the risk of missing eligible articles.

In addition to our systematic search, we manually screened the reference lists of the included studies and relevant review articles to identify any additional studies that may not have been captured initially but could potentially be included in this review.

To formulate the research question and identify the inclusion criteria, we applied the PICOS model (Participants, Interventions, Comparisons, Outcome, and Study Design) [26].

(P) Population: Adults aged 18 years or older hospitalized with uncomplicated pneumonia were eligible for inclusion.

(I) Interventions: physiotherapy interventions.

(C) Comparison: A control intervention in which patients received either a placebo, no treatment, usual care, or standardized conventional care that did not include physiotherapy.

(O) Outcome: Hospital stay and pneumonia-related symptoms.

(S) Study Design: Randomized clinical trials and pilot randomized clinical trials were included.

The exclusion criteria were studies that included physical therapy in the control group; studies involving ICU patients; articles without available full text; and articles written in languages other than English or Spanish.

Following the collection of records from the databases, the study selection process involved eliminating duplicates and screening titles, abstracts, and eligible full texts. To mitigate potential selection bias, two investigators conducted the literature search. Any disagreements were resolved by a third reviewer. After the selection of studies, data extraction and a quality assessment were carried out.

Data extraction was conducted following the data extraction checklist outlined in the Cochrane Handbook for Systematic Reviews [27]. The extracted data encompassed authors, year of publication, study design and setting, treatment status, number of patients, sex distribution, mean age, intervention description, study frequency and duration, and reported outcomes.

The risk of bias of each study was assessed using the Cochrane Risk-of-Bias tool version 2.0 (RoB-2) [28]. The tool evaluates five domains: randomization process, deviations from the intended interventions, missing outcome data, measurement of the outcome, and the selection of the reported result. Studies are categorized based on their risk of bias as high, unclear, or low. The assessment results are generated using the Microsoft Excel spreadsheet template provided with the ROB2 tool.

### 2.3. Methodological Quality Assessment

Outcome quality was assessed using the Downs and Black Checklist [29]. This tool evaluates 27 items across five subscales: reporting, external validity, internal validity, selection bias, and study power. A score of 14 or lower is considered poor quality, a score between 15 and 19 is considered fair quality, a score between 20 and 25 is considered good quality, and a score of 26 or higher is considered excellent quality.

### 2.4. Meta-Analysis

A quantitative analysis was conducted using The Review Manager 5 (Rev-Man version 5.1, updated March 2011) software for all studies presenting post-intervention means and standard deviations of length of stay, exercise capacity at follow-up, and pneumonia-associated symptoms. The mean difference value was used to assess the dyspnea levels, and the length of hospital stay. Data, including final mean values, standard deviations, and the number of patients assessed at different endpoints for each treatment arm, were extracted to estimate overall mean differences between treatment arms.

For articles with insufficient data to calculate effect size (e.g., no provided means or standard deviation), authors were contacted to obtain the required information. When *p*-values or 95% confidence intervals were available, and standard deviations were missing, calculations were performed following the guidelines outlined in the Review Manager manual [27]. These measures were implemented to maximize the reliability and validity of the findings.

Continuous outcomes were analyzed using weighted mean differences (MD). Standardized mean differences were employed when different scales were assumed to measure the same underlying symptom or condition. The 95% confidence intervals (CI) were computed for all outcomes. Overall mean effect sizes were estimated using random effect models or fixed effect models based on I^2^ (I-squared) tests for statistical heterogeneity. I^2^ < 50% is considered to be a meta-analysis with low heterogeneity, and a fixed-effects model was used [27]. A visual inspection of the forest plots for outlier studies was also undertaken. Studies with fewer than 20 participants per group were excluded from the meta-analysis in order to reduce the risk of bias associated with small sample sizes.

## 3. Results

The flow diagram (Figure 1) shows the selection process of the included studies. Finally, nine studies [30,31,32,33,34,35,36,37,38] underwent a full-text screening and were included in the systematic review.

### 3.1. Characteristics of Studies

The characteristics of pneumonia patients (age, sex, pneumonia diagnosis criteria and pneumonia severity) are shown in Table 1. The studies, published between 1978 and 2024, included only randomized control trials. The total sample of patients included in the studies was 1349 hospitalized non-ICU pneumonia patients, with a gender distribution of 56.58% male. The mean age of the patients ranged from 36.80 to 82.5 years.

The pneumonia diagnosis criteria varied between studies. The majority used chest X-rays followed by pneumonia symptoms (cough, fever, pain, dyspnea, etc.) [30,32,33,34,35]. One study used only chest X-rays [31], one study used both chest X-rays and computed tomography followed by pneumonia symptoms [38], and two studies did not specify the diagnostic methods used [36,37]. The pneumonia severity was evaluated with Pneumonia Severity Index (PSI) [34,35], Confusion, Urea, Respiratory rate, Blood pressure, Age ≥ 65 score (CURB-65) [38], and the Simplified Acute Physiology Score (SAPS) [33].

With respect to the methodological quality of the included studies, as assessed by the Downs and Black method, one study was classified as excellent [31], seven studies as good [32,33,34,36,37,38], and two studies as fair [30,35]. Furthermore, the risk of bias in all the studies [30,38] was assessed using the RoB-2 tool (Figure 2), which observed that two of the articles had a high risk of bias [30,31], two had some concerns [34,38], and five had low risk [32,33,35,36,37].

The characteristics of physical therapy interventions for pneumonia patients are shown in Table 2. The most common physical therapy interventions were respiratory physical therapy (RPT) [31], musculoskeletal physical therapy (MPT) [34,35,37,38], and a combination of RPT and MPT [30,36], compared to usual care for hospitalized pneumonia patients, which includes oxygen therapy if needed, intravenous fluids, antibiotics, symptom management, and regular monitoring of vital signs. Additionally, two studies compared RTP and/or MPT with other intervention [32,33].

The timing of the interventions ranged from 10 to 15 min [32,33] to 60 min [36]. Most of the interventions were provided during the hospital stay and every day [30,31,32,33,35,36,37,38]. Furthermore, only two studies evaluated modified the intensity of physical therapy according to a protocol [30,38].

There was variability in the measured outcomes, with the length of hospital stay being the most recorded [30,31,32,33,34,35,36,37,38], as well as the duration of fever [30,31,32], dyspnea [31,36,37], lung volumes and respiratory flows [31], length of intravenous and/or oral antibiotic therapy [31,32,33,35], muscle strength [36], mortality rate [34,35,38], readmission hospital [34,35,38], functional physical capacity [37], general well-being [31].

### 3.2. Results Obtained in Metanalysis

The results obtained in the metanalysis concerning the dyspnea levels are shown in Figure 3. The pooled MD showed significant overall effect of difference RPT and/or MPT compared with usual care (MD = −1.02; 95% CI = −1.78; −0.26; *p* < 0.01). The meta-analysis showed no significant heterogeneity among the included studies (I^2^ = 0%), indicating consistent effect sizes across studies.

The results obtained in the metanalysis concerning the length of hospital stay are shown in Figure 4. The pooled MD showed significant overall effect of difference RPT and/or MPT compared with usual care (MD = −1.47; 95% CI = −1.86; −1.08; *p* < 0.001). The meta-analysis revealed moderate to substantial heterogeneity among the included studies (I^2^ = 63%), suggesting variability in study outcomes that may be attributed to differences in study design, populations, or interventions.

A subgroup analysis was performed. The pooled MD of RPT compared with the usual care subgroup was not estimable due to the small sample size in each group. The pooled MD of MPT compared with usual care subgroup showed significant overall effect of the length of hospital stay (MD = −1.44; 95% CI = −1.85; −1.03; *p* < 0.001). The pooled MD of RPT + MPT compared with the usual care subgroup showed no significant overall effect of the length of hospital stay (MD = −1.10; 95% CI = −3.40; 1.20; *p* = 0.35). The pooled MD of RPT and/or MPT compared with usual care and other intervention subgroup showed significant overall effect of the length of hospital stay (MD = −2.00; 95% CI = −3.49; −0.51; *p* < 0.01).

## 4. Discussion

This systematic review and meta-analysis aimed to evaluate the effectiveness of physiotherapy in patients with non-complicated hospital pneumonia. A synthesis of the results across the included studies reveals consistent evidence that physiotherapy interventions reduce hospital stay and improve dyspnea in hospitalized patients with pneumonia. Despite some variability among study designs and sample sizes, the overall trend supports the clinical benefit of physiotherapy in this population. These outcomes align with the existing literature emphasizing the positive impact of respiratory physiotherapy in pneumonia management [39].

Our review observed the use of PSI, CURB-65 and SAPS to evaluate the pneumonia severity in patients. It is important to note that none of these tools are perfect. A recent meta-analysis compared PSI and CURB-65 in predicting mortality and the need for ICU support, finding that both have strengths and limitations. CURB-65 showed higher sensitivity and specificity for predicting ICU needs, while PSI was more effective in identifying low-risk patients [40,41].

Our study’s findings suggest that physiotherapy interventions in non-ICU hospitalized pneumonia patients significantly reduce hospital stays, highlighting the benefits of physiotherapy in pneumonia management. Additionally, a systematic review [42] found that multimodality respiratory physiotherapy reduced mortality in ICU patients, although it did not have a clear impact on the incidence of ventilator-associated pneumonia or the length of ICU stay. This review, together with our findings, underscores the potential of physiotherapy interventions to enhance patient outcomes in pneumonia patients.

Physiotherapy interventions, including early mobilization and breathing exercises, have been shown to enhance respiratory mechanics and reduce dyspnea. Larsen, T. et al. [39] conducted a systematic review demonstrating that early mobility interventions in pneumonia patients decreased hospital length of stay and improved functional outcomes, which could be associated with alleviated dyspnea. In line with this, our results showed significant differences in dyspnea levels, with better outcomes observed in pneumonia patients who received physical therapy. Further supporting these findings, a pilot study investigated a systematic and progressive early physiotherapeutic mobilization program in patients with community-acquired pneumonia. The study reported a reduction in length of hospital stay in the intervention group compared to usual care. Notably, patients with a greater severity experienced an even greater reduction in length of hospital stay [43].

Our systematic review and meta-analysis found that respiratory physiotherapy has a beneficial effect on dyspnea in patients hospitalized with pneumonia. This finding is consistent with previous evidence supporting the role of physiotherapeutic interventions in improving respiratory symptoms and functional outcomes in acute respiratory conditions. Respiratory physiotherapy reduces respiratory effort, and enhances secretion clearance, improving oxygenation levels and dyspnea scores in patients with pneumonia [44]. Additionally, physiotherapy reduced the incidence of respiratory complications and improved clinical recovery in mechanically ventilated patients with pneumonia [45]. These results support current guidelines recommending physiotherapy as part of multidisciplinary management for hospitalized respiratory patients [13,46].

The evidence [47] supports the integration of physiotherapy into hospitalized pneumonia care. Despite the observed benefits, it is important to note that the effectiveness of physiotherapy may vary according to factors such as pneumonia severity, patient comorbidities, and the timing and intensity of intervention initiation.

Several limitations of this systematic review and meta-analysis are important. First, a limitation is the small number of randomized controlled trials included, which may reduce the overall comprehensiveness of the evidence and limit the generalizability of the findings. Second, the methodological quality of the included studies was suboptimal, which may have influenced the outcomes of the analysis. In addition, there was considerable heterogeneity in the overall results. Additionally, this meta-analysis focused on short-term effects due to the absence of follow-up data in the included studies. Furthermore, some studies were limited by relatively small sample sizes, which may have affected the reliability and generalizability of their reported outcomes. In addition, some included studies did not assess pneumonia severity, either due to publication prior to the adoption of PSI and CURB-65, or because their primary focus was not severity stratification. Lastly, several studies were excluded from the meta-analysis as they lacked the requisite data for comparison and analysis with the other studies.

## 5. Conclusions

Our study highlights the important role of physiotherapy in the management of pneumonia patients hospitalized in non-ICU settings. This finding demonstrates that physiotherapy interventions have the potential to effectively reduce the length of hospital stay, which can alleviate healthcare burdens and improve resource utilization. Additionally, physiotherapy supports overall clinical recovery by enhancing respiratory function and physical capacity. Thus, our results also highlight a notable reduction in dyspnea levels among these patients, contributing to improved patient comfort and quality of life during hospitalization. Nevertheless, due to the variability in clinical practices, additional high-quality research is essential to confirm these findings, assess their long-term impact, and develop standardized, evidence-based treatment protocols adapted to diverse healthcare settings.

## Figures and Tables

**Figure 1 healthcare-13-01444-f001:**
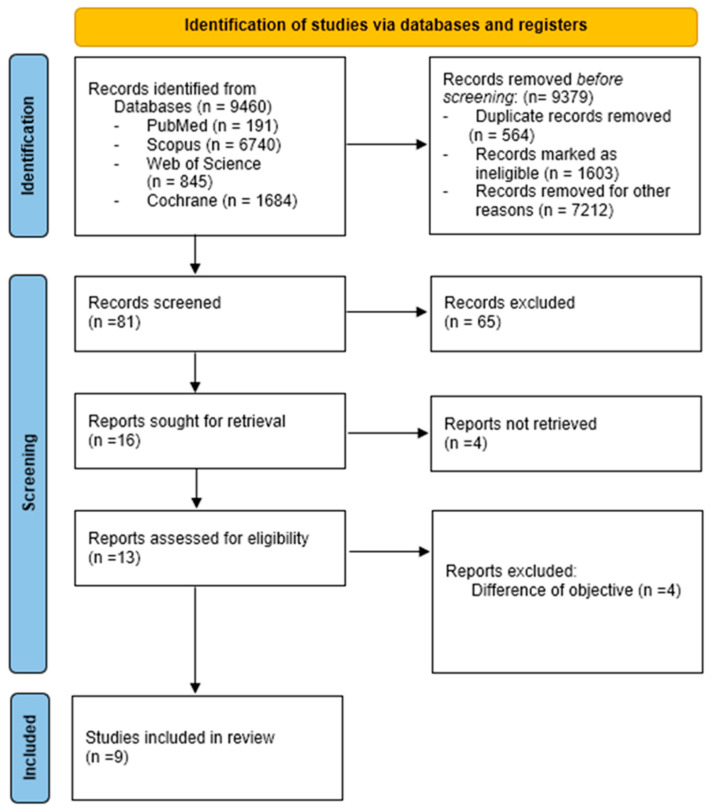
PRISMA flow diagram.

**Figure 2 healthcare-13-01444-f002:**
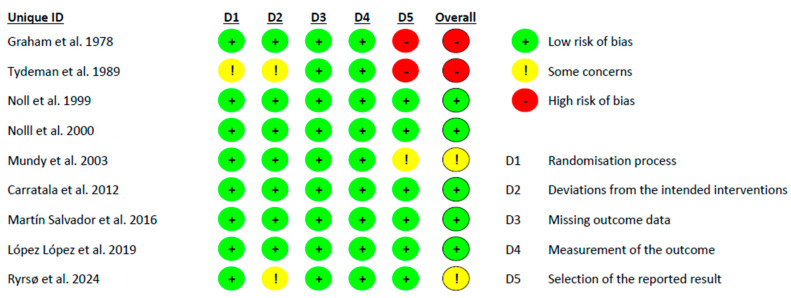
Risk of bias in included studies [30,31,32,33,34,35,36,37,38].

**Figure 3 healthcare-13-01444-f003:**
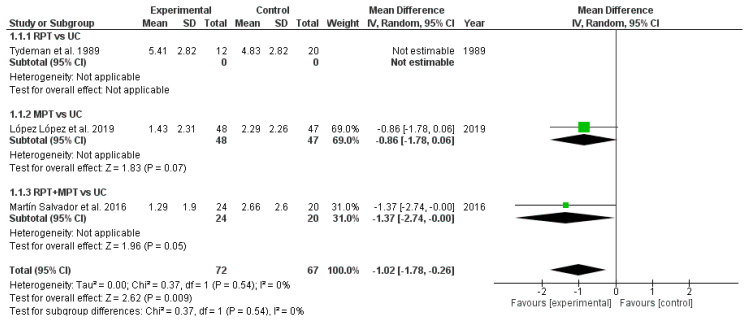
Meta-analysis of dyspnea levels. (UC: usual care; RPT: respiratory physical therapy; MPT: musculoskeletal physical therapy; SD: standard deviation; IV: inverse variance; CI: confidence interval.) [31,36,37].

**Figure 4 healthcare-13-01444-f004:**
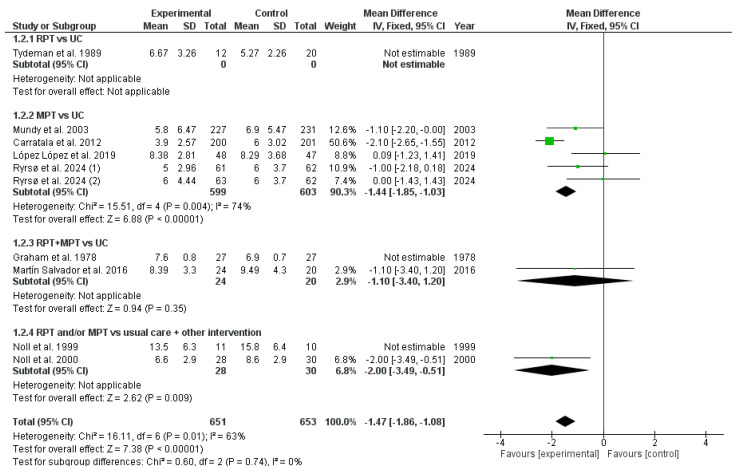
Meta-analysis of length of hospital stay. (UC: usual care; RPT: respiratory physical therapy; MPT: musculoskeletal physical therapy; SD: standard deviation; IV: inverse variance; CI: confidence interval.) [30,31,32,33,34,35,36,37,38].

**Table 1 healthcare-13-01444-t001:** Characteristics of the included studies.

Study	Sample Size (% Male)	Sample Age (Mean Years ± SD)	Pneumonia Diagnosis Criteria	Pneumonia Severity	Downs and Black Score
Graham et al. 1978 [30].	IG: 27 (51.85) CG: 27 (48.14)	IG: 63 ± 20.78 CG: 61 ± 15.58	CXR and the onset of typical symptoms (coughing and fever).	NR	18
Tydeman et al. 1989 [31].	IG: 12 (NR) CG: 20 (NR)	IG: 42.08 ± 15.59 CG: 36.80 ± 13.91	CXR	NR	26
Noll et al. 1999 [32].	IG: 11 (27.27) CG: 10 (30)	IG: 78.7 ± NR CG: 82.5 ± NR	CXR and at least two clinical findings (fever, leukocytosis, cough, or acute mental status changes).	NR	20
Noll et al. 2000 [33].	IG: 28 (50) CG: 30 (53.3)	IG: 77 ± 17.2 CG: 77.7 ± 17.1	CXR and at least two clinical findings (fever, leukocytosis, new cough, or acute mental status changes).	SAPS, mean ± SD. IG:9.4 ± 4.0 CG:9.6 ± 3.5	22
Mundy et al. 2003 [34].	IG: 227 (44) CG: 231 (44)	NR	CXR and one major criteria (cough, sputum production, or temperature >37.8 °C) or two minor criteria (pleuritic chest pain, dyspnea, altered mental status, pulmonary consolidation on examination, or leukocyte count >12,000/μL).	PSI, n(%). IG: Low: 129(57); Moderate: 71(31); Hight:27(12). CG: Low: 123(53); Moderate: 65(28), High:43(19).	25
Carratala et al. 2012 [35].	IG: 200 (66) CG: 201 (64.2)	IG: 71.5 ± 14 CG: 69.7 ± 15.1	CXR and following symptoms: fever or hypothermia, new cough with or without sputum production, pleuritic chest pain, dyspnea, and altered breath sounds on auscultation.	PSI, mean ± SD. IG:100.5 ± 32.5 CG:101.1 ± 31.5	17
Martín-Salvador et al. 2016 [36].	IG: 24(83.2) CG: 20(78)	IG: 78.82 ± 6.3 CG: 77.40 ± 5.2	NR	NR	23
López-López et al. 2019 [37].	IG: 48(41.67) CG: 47(57.45)	IG: 74.92 ± 11.03 CG: 72.53 ± 9.24	NR	NR	25
Ryrsø et al. 2024 [38].	IG1: 61 (49) IG2: 63 (37) CG: 62 (48)	IG1: 70 ± 14 IG2: 69 ± 14 CG: 68 ± 13	CXR or computed tomography scan with one symptom (cough, chest pain, fever, hypothermia or dyspnea).	CURB-65, n(%). IG1: 0-1: 35(57); 2: 21(34); 3-5: 5(8). IG2: 0-1: 36(57); 2: 22(35); 3-5: 5(8). CG: 0-1: 36(58); 2: 21(34); 3-5: 5(8).	22

NR: Not reported; IG: intervention group; CG: control group; PSI: Pneumonia Severity Index; CURB-65: Confusion, Urea, Respiratory rate, Blood pressure, Age ≥ 65 score; SAPS: the simplified acute physiology score; CXR: chest x-ray.

**Table 2 healthcare-13-01444-t002:** Characteristics of the physical therapy interventions for pneumonia patients.

Study	Interventions	PT Program Description	Intervention Timing, Duration, and Frequency and Intensity	Outcomes Measures and Results	Conclusion
Graham et al. 1978 [30].	PT program: RPT + MPT + UC CG: UC	Chest physiotherapy + positive pressure breathing + UC.	20 min, until discharge (average 5 days), every day, and 15 cm of water end-inspiratory pressure.	Duration of fever, IG > CG, *p* = 0.64; LOS, IG > CG, *p* = 0.52; Discharge chest x-ray, *p* = 0.79.	Pneumonia does not resolve faster with chest physiotherapy, and intermittent positive pressure breathing. NOT EFFECTIVE
Tydeman et al. 1989 [31].	PT program: RPT CG: UC	Breathing control, localized expansion, postural drainage, thoracic expansion with vibrations, and percussion.	15–20 min twice a day, until discharge, every day and NR.	LOS, IG < CG, *p* > 0.05; total time antibiotics, *p* > 0.05; predicted FVC, IG > CG, *p* > 0.05, dyspnea IG > CG, *p* > 0.05, general well-being, *p* > 0.05; Presence of air bronchogram and pleural effusion; duration of sputum production and weight, *p* > 0.05.	Physical therapy remains unproven for the treatment of previously fit patients with uncomplicated pneumonia. NOT EFFECTIVE
Noll et al. 1999 [32].	PT program: RPT + MPT CG: Light touch treatment + UC	OMT: paraspinal muscle inhibition, rib raising, diaphragmatic release, condylar decompression, cervical soft tissue, myofascial release to thoracic inlet, and thoracic lymphatic pump.	10–15 min, twice a day, until discharge, every day, and NR.	Duration of fever, IG > CG, *p* > 0.05; duration of leukocytosis, IG < CG, *p* > 0.05; LOS, IG < CG, *p* > 0.05; length of IV and oral antibiotic therapy, IG < CG, *p* > 0.05; length of IV antibiotic therapy, IG < CG, *p* > 0.05; length of oral antibiotic therapy, IG > CG, *p* = 0.04.	Although the mean duration of leukocytosis, intravenous antibiotic treatment, and length of stay were shorter in the adjunctive osteopathic manipulative treatment group, these measures did not reach statistical significance. However, the mean duration of oral antibiotic use reached statistical significance at 3.1 days in the treatment group and 0.8 days in the control group. NOT EFFECTIVE
Noll et al. 2000 [33].	PT program: RPT + MPT CG: Light touch treatment + UC	OMT: paraspinal muscle inhibition, rib raising, diaphragmatic release, condylar decompression, cervical soft tissue, myofascial release to thoracic inlet, and thoracic lymphatic pump.	10–15 min, twice a day, until discharge, every day, and NR.	LOS, IG < CG, *p* = 0.014; length of IV and oral antibiotic therapy, IG < CG, *p* = 0.003; length of IV antibiotic therapy, IG < CG, *p* = 0.002; length of oral antibiotic therapy, *p* = 0.952; febrile shifts, *p* = 0.738; tachypnea shifts, *p* = 0.673; tachycardia shifts, *p* = 0.927.	The osteopathic manipulative treatment group had a significantly shorter duration of intravenous antibiotic treatment, and a shorter hospital stay. EFFECTIVE
Mundy et al. 2003 [34].	PT program: MPT CG: UC	EM	20 min, every day, NR and NR.	LOS, IG < CG; Mortality rate, IG < CG; hospital charges, IG < CG, *p* = 0.05; hospital readmissions, *p* > 0.05; emergency department, *p* > 0.05; chest radiographs, *p* > 0.05.	EM of hospitalized adults reduces hospital length of stay. EM was defined as sitting or walking out of bed for at least 20 min during the first 24 h of hospitalization. Progressive mobilization occurred each subsequent day during hospitalization EM was effective. EFFECTIVE
Carratala et al. 2012 [35].	PT program: MPT CG: UC	EM, objective criteria for oral antibiotics, predefined discharge criteria.	20 min, every day, until discharge, and NR.	LOS, IG < CG, *p* < 0.001; length of IV antibiotic therapy, IG < CG, *p* < 0.001; adverse drug reactions, IG < CG, *p* < 0.001; medical complications, IG < CG, *p* = 0.340; Mortality rate, IG > CG, *p* = 0.450; hospital readmissions, IG > CG, *p* = 0.590.	Three-step critical pathway including early mobilization and use of objective criteria for switching to oral antibiotic therapy and for deciding on hospital discharge or usual care was safe and effective in reducing the duration of intravenous antibiotic therapy and LOS for CAP and did not adversely affect patient outcomes. EFFECTIVE
Martín-Salvador et al. 2016 [36].	PT program: RPT + MPT + UC CG: UC	Ventilatory re-education, neuromuscular electrical stimulation with exercises, and resistance elastic band exercises + UC.	60 min, every day, until discharge and NR.	LOS, IG < CG, *p* = 0.437; dyspnea, IG < CG, *p* = 0.041; muscle strength, IG < CG, *p* < 0.05; functional capacity related respiratory symptoms, IG > CG, *p* = 0.277.	Between-group analysis showed that after the physical therapy intervention (experimental vs. control) significant differences were found in perceived dyspnea (*p* = 0.041), and right and left quadriceps muscle strength (*p* = 0.008 and *p* = 0.010, respectively). In addition, the subscale of “domestic activities” of the functional ability related to respiratory symptoms questionnaire showed significant differences (*p* = 0.036). EFFECTIVE
López-López et al. 2019 [37].	PT program: MPT + UC CG: UC	Warm-up, neuromuscular electrical stimulation with exercises, and resistance elastic band exercises, cool-down + UC.	45 min, every day, until discharge and NR.	LOS, IG < CG, *p* = 0.897; dyspnea, IG < CG, *p* = 0.101; SPPB, IG > CG, *p* = 0.027; Fatigue, IG < CG, *p* = 0.030; LCQ, IG > CG, *p* = 0.041.	Fatigue (32.04 (18.58) vs. 46.22 (8.90)) and cough (18.84 (2.47) vs. 17.40 (3.67)) showed higher improvement in the intervention group, and significant differences were observed between the groups. An integrated program of physical and electrical stimulation during hospitalization improves physical and functional performance in patients with pneumonia. EFFECTIVE
Ryrsø et al. 2024 [38].	PT program1: MPT + UC PT program2: MPT + UC CG: UC	PT program1: early mobilization with in-bed cycling intervention and booklet exercise + UC.PT program2: The booklet intervention included bodyweight resistance and walking exercises + UC.	PT program1:30 min, every day, until discharge and Based on SpO2, HR, and Borg, gradually increased. PT program2: 30 min, every day, until discharge and the exercise instructor gradually increased.	LOS, IG1 < (CG ≈ IG2); mortality rate, IG2 < (CG ≈ IG1); Hospital readmission, IG1 < IG2 < CG.	Supervised in-bed cycling and booklet exercise had no effect on LOS. Similarly, exercise training had no statistically significant effect on 90-day readmission risk. NOT EFFECTIVE

NR: not reported; EM: early mobilization; PT program: physical therapy program; UC: usual care; RPT: respiratory physical therapy; MPT: musculoskeletal physical therapy; PM: pharmaceutical management; LOS: length of hospital stay; FVC: forced vital capacity; FEV1: forced expiratory volume in 1 s; OMT: osteopathic manipulative treatment; IV: intravenous; SPPB: Short Physical Performance Battery; LCQ: Leicester cough questionnaire; SpO2: oxygen saturation; HR: heart rate.

## Data Availability

No new data were created or analyzed in this study.

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
