# Peer review of "Effectiveness of Physical and Therapy Interventions for Non-ICU Hospitalized Pneumonia Patients: A Systematic Review of Randomized Controlled Trials"

_healthcare, 2025, doi:10.3390/healthcare13121444_

Round 1
Reviewer 1 Report (Previous Reviewer 1)
Comments and Suggestions for Authors
Thank you for revising the manuscript according to the reviewers' comments.
In general, the results of meta-analyses are less reliable when papers with small sample sizes and low levels of evidence are included.
Therefore, although fewer papers will be included in the analysis, I believe it is better to include papers with large sample sizes and clear evidence of the effectiveness of physical therapy.
This approach will clarify the clinical significance of physical therapy, which is the aim of this paper. This approach will make this study more reliable.
I hope these review comments are helpful.
Line 76:
I think “Thus” or “Therefore” would be more appropriate for this expression (Despite the potential benefits of physiotherapy).
Line 81-82:
It can be changed as follows:
This systematic review and meta-analysis aimed to evaluate the effectiveness of physiotherapy in treating uncomplicated hospital-acquired pneumonia.
Please match Figure 2 with the corresponding explanatory text.
For example, Graham et al. 1978 [30], Tydeman et al. 1989 [31] in the figure 3.
The figure 3 should be presented/made in such a way that readers can understand the analysis results from the figures.
FIGURES 3 and 4
In meta-analysis, excluding studies with high risk of bias is common practice to improve the reliability of the results.
Including such studies without appropriate handling can lead to unreliable and misleading conclusions.
Therefore, the meta-analysis results in Figures 3 and 4, I believe that Graham et al. 1978 [30], Tydeman et al. 1989 [31], and Noll et al. 1999 [32] (with very few samples) results should be excluded.
Additionally, please reconsider the exclusion criteria.
TABLE 2
Please confirm the probability {p=277} in Table 2 by Martín-Salvador et al. 2016 [36]. 
Is the "electrical therapy" in Table 2 mentioned in López-López et al., 2019, the same as "neuromuscular electrical stimulation"? Please check the paper carefully and clarify this point. The term "electrical therapy" is also used in fields other than physical therapy; thus, the specific treatment content cannot be determined.
Line 213-214:
There was multiple variability in outcomes measured, the length of hospital stay <de> most recorded in each one [30-38]. <Aso>, duration of fever [30-32], dyspnea [31,36,37], lung
Does this sentence mean the following? Please check it and correct any errors.
There was variability in the measured outcomes, with length of hospital stay being the most recorded [30-38]. Also, the duration of fever [30-32] and dyspnea [31, 36, 37] ...
Line 280-282:
Authors can make your writing more persuasive by editing it as follows:
The evidence [INSERT references number] supports the integration of physiotherapy into hospitalized pneumonia care. However, the effectiveness of physiotherapy may vary according to factors such as pneumonia severity, patient comorbidities, and the timing of intervention initiation.
Please explain the abbreviations I2, MD, and CI.
As mentioned above, the English text also contains many errors. It should be proofread by an English expert.
I hope this helps.
Comments on the Quality of English LanguageEnglish text also contains many errors. It should be proofread by an English expert.
Author Response
Dear Reviewer 1,
Point-by-point reply to the reviewer's comments:
Comment 1. Thank you for revising the manuscript according to the reviewers' comments.
In general, the results of meta-analyses are less reliable when papers with small sample sizes and low levels of evidence are included.
Therefore, although fewer papers will be included in the analysis, I believe it is better to include papers with large sample sizes and clear evidence of the effectiveness of physical therapy.
This approach will clarify the clinical significance of physical therapy, which is the aim of this paper. This approach will make this study more reliable.
I hope these review comments are helpful.
Response 1: Thank you for your insightful comment regarding the inclusion of studies with small sample sizes and lower levels of evidence. We fully agree that incorporating studies with larger sample sizes and higher methodological quality would increase the overall reliability and clinical significance of the meta-analysis.
However, due to the specific focus and strict inclusion criteria of our methodology, we found that most of the available studies meeting our topic criteria were characterized by small sample sizes and varied levels of evidence. Limiting the selection further would have significantly reduced the number of eligible studies and may have compromised the ability to conduct a meaningful analysis.
In response to your comment, we have now explicitly acknowledged this issue as a limitation in the discussion section, emphasizing that the nature of the available evidence in this field currently poses constraints to the generalizability and strength of our conclusions.
Comment 2. Line 76: I think “Thus” or “Therefore” would be more appropriate for this expression (Despite the potential benefits of physiotherapy).
Response 2. We agree that "Thus" or "Therefore" would improve the clarity and flow of the sentence in Line 76. We have revised the sentence accordingly to reflect this recommendation
Comment 3. Line 81-82: It can be changed as follows: This systematic review and meta-analysis aimed to evaluate the effectiveness of physiotherapy in treating uncomplicated hospital-acquired pneumonia.
Response 3. Thank you for your suggestion regarding the wording of Lines 81–82. We appreciate your recommendation for improving the clarity and precision of the objective statement. We believe this phrasing improves the overall readability of the manuscript and better reflects the aim of our study.
Comment 4: Please match Figure 2 with the corresponding explanatory text. For example, Graham et al. 1978 [30], Tydeman et al. 1989 [31] in the figure 3.
Response 4: Thank you for pointing this out. We have clarified which studies are represented in Figure 2 to avoid any confusion. We appreciate your observation, which helped improve the clarity and consistency of our manuscript.
Comment 5: The figure 3 should be presented/made in such a way that readers can understand the analysis results from the figures.
Response 5: Thank you for your valuable feedback regarding Figure 3. We agree that figures should clearly convey the analysis results to the reader. In response, we have revised Figure 3 and 4 to enhance its clarity and readability. We added a more informative caption to ensure that the results of the analysis are more easily understood.
Comment 6: FIGURES 3 and 4. In meta-analysis, excluding studies with high risk of bias is common practice to improve the reliability of the results.
Including such studies without appropriate handling can lead to unreliable and misleading conclusions. Therefore, the meta-analysis results in Figures 3 and 4, I believe that Graham et al. 1978 [30], Tydeman et al. 1989 [31], and Noll et al. 1999 [32] (with very few samples) results should be excluded. Additionally, please reconsider the exclusion criteria.
Response 6: Thank you for your insightful comment regarding the inclusion of studies with a high risk of bias and very small sample sizes in the meta-analysis (Graham et al. 1978 [30], Tydeman et al. 1989 [31], and Noll et al. 1999 [32]). In response to your suggestion, we have removed these studies from the analysis due to their small sample sizes, which we now recognize as a potential source of bias. We have also updated the exclusion criteria accordingly in the Methods section to reflect this change.
As a result, we have re-run the meta-analysis, and the revised results are now presented in the updated versions of Figures 3 and 4. The manuscript has been modified to incorporate these new findings. We sincerely appreciate your recommendation, which has helped improve the robustness and reliability of our study.
Comment 7. TABLE 2 Please confirm the probability {p=277} in Table 2 by Martín-Salvador et al. 2016 [36]. Is the "electrical therapy" in Table 2 mentioned in López-López et al., 2019, the same as "neuromuscular electrical stimulation"? Please check the paper carefully and clarify this point. The term "electrical therapy" is also used in fields other than physical therapy; thus, the specific treatment content cannot be determined.
Response 7. Thank you for your careful observation regarding Table 2. We have reviewed the original study by Martín-Salvador et al. (2016) and confirm that the reported p-value was indeed p = 0.277. The table has been checked and remains accurate. Regarding the term "electrical therapy" as mentioned in López-López et al. (2019), we agree that this terminology may be ambiguous. After carefully reviewing the original article, we confirm that the intervention referred to as "electrical therapy" corresponds specifically to neuromuscular electrical stimulation (NMES). We have now revised Table 2 to reflect this more precise terminology and avoid confusion in both studies. We appreciate your comment, which helped us improve the accuracy and clarity of the intervention descriptions in the manuscript.
Comment 8: Line 213-214: There was multiple variability in outcomes measured, the length of hospital stay <de> most recorded in each one [30-38]. <Aso>, duration of fever [30-32], dyspnea [31,36,37], lung.
Does this sentence mean the following? Please check it and correct any errors.
There was variability in the measured outcomes, with length of hospital stay being the most recorded [30-38]. Also, the duration of fever [30-32] and dyspnea [31, 36, 37] ...
Response 8: Thank you for your observation and for suggesting a clearer version of the sentence. You are correct in your interpretation. We have revised the sentence accordingly to improve clarity and readability.
Comment 9: Line 280-282:
Authors can make your writing more persuasive by editing it as follows:
The evidence [INSERT references number] supports the integration of physiotherapy into hospitalized pneumonia care. However, the effectiveness of physiotherapy may vary according to factors such as pneumonia severity, patient comorbidities, and the timing of intervention initiation.
Response 9: Thank you for your helpful suggestion to improve the clarity and persuasiveness of our writing. We have revised the text as recommended.
Comment 10: Please explain the abbreviations I2, MD, and CI.
Response 10: Thank you for pointing this out. We have now added explanations for the abbreviations used in the meta-analysis results to enhance clarity for all readers. Specifically:
- I² stands for I-squared.
- MD refers to Mean Difference.
- CI stands for Confidence Interval
These explanations have been added to the Methods section.
Comment 11. Comments on the Quality of English Language. English text also contains many errors. It should be proofread by an English expert.
Response 11. Thank you for your comment regarding the quality of the English language. We acknowledge the need for improvement in clarity and grammar. In response, the manuscript has been thoroughly revised. We believe the revised version now meets the standards required for publication.
Reviewer 2 Report (Previous Reviewer 2)
Comments and Suggestions for Authors
The manuscript entitled “Effectiveness of physical y therapy interventions for non-ICU hospitalized pneumonia patients: A systematic review of randomized controlled trials” was interesting. The authors aimed to evaluate the efficacy of physical therapy in impacting health-related outcomes in hospitalized non-ICU pneumonia. However, the following issues need further attentions:
- I think there is a typo “y letter” in the title.
- There are some highlights in the manuscript. I am not sure whether this is an original submission or a revised one.
- Similar studies should be reviewed in the introduction section.
- The databases mentioned in the abstract are different from the databases in the PRISMA diagram.
- The initial search results were quite a lot and only 9 studies were selected. This suggested that the search strings were not specific enough.
- Search strategies were not available to me to examine them in different databases.
- A synthesis section should be added to the end of the results section.
- The conclusion section needs to be expanded to present the main findings of the research.
Author Response
Dear Reviewer 2,
Point-by-point reply to the reviewer's comments:
The manuscript entitled “Effectiveness of physical y therapy interventions for non-ICU hospitalized pneumonia patients: A systematic review of randomized controlled trials” was interesting. The authors aimed to evaluate the efficacy of physical therapy in impacting health-related outcomes in hospitalized non-ICU pneumonia. However, the following issues need further attentions:
Comment 1: I think there is a typo “y letter” in the title.
Response 1: Thank you for bringing this to our attention. We have identified and corrected the typographical error in the title. We appreciate your careful review.
Comment 2: There are some highlights in the manuscript. I am not sure whether this is an original submission or a revised one.
Response 2: Thank you for your observation. This is a revised version of the manuscript, submitted in response to the previous round of reviewer comments. The highlights were included to clearly indicate the changes made throughout the text. We will remove the highlights in the final version.
Comment 3: Similar studies should be reviewed in the introduction section.
Response 3: Thank you for your comment. In response, we have reviewed and incorporated relevant similar studies into the Introduction section to provide a more comprehensive background and to better position our study within the existing literature. These additions help clarify the current knowledge on the topic and highlight the specific contribution of our work.
Comment 4: The databases mentioned in the abstract are different from the databases in the PRISMA diagram.
Response 4: Thank you for your observation. As you indicated, there was an inconsistency between the databases listed in the abstract, methods section, and those shown in the PRISMA flow diagram. We have now carefully reviewed and corrected to match the databases actually used and presented in the PRISMA diagram.
Comment 5: The initial search results were quite a lot and only 9 studies were selected. This suggested that the search strings were not specific enough.
Response 5: Thank you for your valuable observation. We acknowledge that the initial search yielded a large number of studies, from which only 9 met the inclusion criteria. This is primarily due to our decision to use broad and sensitive search strings to ensure that all potentially relevant studies were captured, minimizing the risk of omitting eligible articles. While this approach may reduce specificity, it is a common and deliberate strategy in systematic reviews to enhance comprehensiveness. We have now clarified this point in the Methods section.
Comment 6: Search strategies were not available to me to examine them in different databases.
Response 6: Thank you for your comment. The detailed search strategies for each database were included as supplementary material with the submission. We apologize if this was not clearly indicated.
Comment 7: A synthesis section should be added to the end of the results section.
Response 7: Thank you for your valuable suggestion. We have incorporated a synthesis summary of the main findings; however, we placed it at the beginning of the Discussion section rather than at the end of the Results section. We believe this placement allows for a better transition from presenting the data to interpreting its significance. We hope this adjustment meets your expectations.
Comment 8: The conclusion section needs to be expanded to present the main findings of the research.
Response 8: Thank you for your valuable feedback. In response, the revised conclusion now clearly summarizes the key results and their implications for clinical practice, providing a stronger and more informative closing to the manuscript.
Reviewer 3 Report (Previous Reviewer 3)
Comments and Suggestions for Authors
The reviewer not only offered a thorough critique but also suggested a substantially improved revision of the manuscript, even though small sample size of the systematic review.
Author Response
Dear Reviewer 3,
Point-by-point reply to the reviewer's comments:
Comment 1. The reviewer not only offered a thorough critique but also suggested a substantially improved revision of the manuscript, even though small sample size of the systematic review.
Response 1. Thank you very much for your thorough and constructive review. We sincerely appreciate the time and effort you dedicated to evaluating our manuscript and for providing thoughtful suggestions that have significantly improved the quality and clarity of our work. Although the sample size in the systematic review was limited, reviewer feedback allowed us to strengthen the methodology and the results. We are grateful for your valuable contribution to enhancing this study.
Round 2
Reviewer 1 Report (Previous Reviewer 1)
Comments and Suggestions for Authors
Thank you so much for revising your manuscript.
To further improve the quality of your paper, please complete the following tasks.
The abbreviation "ICU" is not explained anywhere in the paper. Please check and correct this.
The significance level of the statistical analysis is
P = 0.009, which can be corrected to p < 0.01; p < 0.00001, which can be corrected to p < 0.001. Please check the entire paper and make the necessary corrections.
Line 76: Chen, X., et al. (2022) [23] observed the effect of chest physio...
This journal does not use APA style.
Please revise as follows:
"Chen, X., et al. [23], observed the effect of chest physiotherapy..."
Figure 2: Risk of Bias in Included Studies.
Revise "Low risk" and "High risk" to "Low risk of bias" and "High risk of bias," respectively.
Table 1: Characteristics of the Included Studies
There are many places where periods are missing in the explanatory text. Please check and revise them.
Figures 3 and 4: Add explanations of the abbreviations SD, IV, and CI below the figures.
I hope this is helpful.
Author Response
Thank you so much for revising your manuscript.
To further improve the quality of your paper, please complete the following tasks.
Comment 1. The abbreviation "ICU" is not explained anywhere in the paper. Please check and correct this.
Response 1. Thank you for your comment. We have addressed this issue by defining the abbreviation "ICU" (Intensive Care Unit) at its first occurrence in the manuscript (Line 21 and 77). We appreciate your careful review
Comment 2. The significance level of the statistical analysis is
P = 0.009, which can be corrected to p < 0.01; p < 0.00001, which can be corrected to p < 0.001. Please check the entire paper and make the necessary corrections.
Response 2. Thank you for your valuable feedback. We have carefully reviewed the manuscript and corrected the significance levels as suggested, ensuring that:
- *P = 0.009* is now presented as p < 0.01
- p < 0.00001 is now presented as p < 0.001
All instances throughout the paper have been updated accordingly. We appreciate your attention to detail.
Comment 3. Line 76: Chen, X., et al. (2022) [23] observed the effect of chest physio...
This journal does not use APA style.
Please revise as follows:
"Chen, X., et al. [23], observed the effect of chest physiotherapy..."
Response 3. Thank you for your observation. We have revised the citation format on line 76, and throughout the manuscript where necessary, to align with the journal’s required referencing style. We appreciate your attention to detail
Comment 4. Figure 2: Risk of Bias in Included Studies.
Revise "Low risk" and "High risk" to "Low risk of bias" and "High risk of bias," respectively.
Response 4. Thank you for your comment. We have revised the labels in Figure 2 as suggested. "Low risk" and "High risk" have been updated to "Low risk of bias" and "High risk of bias," respectively, to improve clarity and align with standard terminology.
Comment 5. Table 1: Characteristics of the Included Studies
There are many places where periods are missing in the explanatory text. Please check and revise them.
Response 5. Thank you for your comment. We have carefully reviewed the explanatory text in Table 1 and 2 and have added the missing periods where necessary, particularly in the definitions of abbreviations and other descriptive notes, to ensure correct punctuation and consistency throughout.
Comment 6. Figures 3 and 4: Add explanations of the abbreviations SD, IV, and CI below the figures.
Response 6. Thank you for your helpful suggestion. We have added explanations of the abbreviations SD (Standard Deviation), IV (Inverse Variance), and CI (Confidence Interval) below Figures 3 and 4 to improve clarity for the reader.
Reviewer 2 Report (Previous Reviewer 2)
Comments and Suggestions for Authors
I appreciate the authors for their time and efforts to revise the manuscript. However, as search strategies were not included in the manuscript, I couldn't check them. Therefore, the comment is still remaining: "The initial search results were quite a lot and only 9 studies were selected. This suggested that the search strings were not specific enough". Moreover, discussion section seems to be brief and needs to be expanded.
Author Response
Comment 1. I appreciate the authors for their time and efforts to revise the manuscript. However, as search strategies were not included in the manuscript, I couldn't check them. Therefore, the comment is still remaining: "The initial search results were quite a lot and only 9 studies were selected. This suggested that the search strings were not specific enough". Moreover, discussion section seems to be brief and needs to be expanded.
Response 1. Thank you very much for your continued feedback and thoughtful comments.
Regarding the search strategies, we acknowledge your concern. Due to the extensive length and complexity of the search strings used across multiple databases, we have included the full search strategies as a supplementary file rather than in the main text, to maintain clarity and readability in the manuscript. We have provided clarification in the Methods section and have included the supplementary file again in the current revision.
As for the discussion section, we agree with your observation. We have expanded the discussion to include a more in-depth analysis of the main findings. We believe these additions provide a more comprehensive context for interpreting the results.
This manuscript is a resubmission of an earlier submission. The following is a list of the peer review reports and author responses from that submission.
Round 1
Reviewer 1 Report
Comments and Suggestions for Authors
This study aimed to determine the effectiveness of physiotherapy in patients with uncomplicated hospital-acquired pneumonia through a systematic review and meta-analysis. The systematic review and meta-analysis methods were conducted appropriately and are evaluated to be very good research procedure.
The reviewer read the articles selected by the authors and extracted the conclusions into a table below.
The authors formulated the research question using the PICOS model.
There are five papers that do not specify the severity of pneumonia in the target population.
For physical therapy for pneumonia, the interventions that were ineffective were chest physiotherapy, physical therapy, adjunctive osteopathic manipulative treatment, supervised in-bed cycling, and booklet exercise.
On the other hand, the following interventions were effective: osteopathic manipulative treatment, early mobilization, physical therapy intervention (breathing exercises, electrostimulation, exercises with elastic bands, and relaxation), supervised in-bed cycling, and book exercise among the selected papers.
Although the selected studies were randomized clinical trials, two studies with very small sample sizes were included.
However, I believe that the conceptualization of physical therapy for uncomplicated hospital-acquired pneumonia, which is the focus of this study, has not been adequately established extracted papers result and conclusion.
Therefore, I believe that conclusions regarding the objectives of this study were not reached.
"What kind of intervention is physical therapy for pneumonia?"
By setting this research question, I believe this study will be able to produce optimal results.
This study is an important study to clarify the clinical value and importance of physical therapy. I recommend conducting a thorough systematic review and working to establish a conceptual framework for physiotherapy in uncomplicated hospital-acquired pneumonia.
Reference paper No. 34 is available in English. Please cite the English version as a reference.
I hope this helps.
Study |
Sample Size (% male) |
Pneumonia severity |
Results |
Graham et al. 1978 [28] |
IG: 27 (51.85) CG: 27 (48.14) |
NR |
Pneumonia does not resolve faster with chest physiotherapy and intermittent positive pressure breathing. NOT EFFECTIVE |
Tydeman et al.1989 [29] |
IG: 12 (NR) CG: 20 (NR) |
NR |
Physical therapy remains unproven for the treatment of previously fit patients with uncomplicated pneumonia. NOT EFFECTIVE |
Noll et al.1999 [30] |
IG:11 (27.27) CG:10 (30) |
NR |
Although the mean duration of leukocytosis, intravenous antibiotic treatment, and length of stay were shorter in the adjunctive osteopathic manipulative treatment group, these measures did not reach statistical significance. However, the mean duration of oral antibiotic use reached statistical significance at 3.1 days in the treatment group and 0.8 days in the control group. NOT EFFECTIVE |
Noll et al. 2000 [31] |
IG:28 (50) CG:30 (53.3) |
SAPS, mean±SD IG:9.4±4.0 CG:9.6±3.5 |
The osteopathic manipulative treatment group had a significantly shorter duration of intravenous antibiotic treatment and a shorter hospital stay. EFFECTIVE |
Mundy et al. 2003 [32] |
IG:227 (44) CG:231 (44) |
PSI, n(%) IG: Low: 129(57); Moderate: 71(31); Hight:27(12) CG: Low: 123(53); Moderate: 65(28), High:43(19) |
Early mobilization (EM) of hospitalized adults with community-acquired pneumonia (CAP) reduces hospital length of stay. EM was defined as sitting or walking out of bed for at least 20 minutes during the first 24 hours of hospitalization. Progressive mobilization occurred each subsequent day during hospitalization EM was effective. EFFECTIVE |
Carratala et al. 2012 [33] |
IG:200 (66) CG:201(64.2) |
PSI, mean±SD IG:100.5±32.5 CG:101.1±31.5 |
3-step critical pathway including early mobilization and use of objective criteria for switching to oral antibiotic therapy and for deciding on hospital discharge or usual care was safe and effective in reducing the duration of intravenous antibiotic therapy and LOS for CAP and did not adversely affect patient outcomes. EFFECTIVE |
Martín-Sal-vador et al. 2016 [34] |
IG:24 (83.2) CG:20 (78) |
NR |
Between-groups analysis showed that after the physical therapy intervention (breathing exercises, electrostimulation, exercises with elastic bands and relaxation) (experimental vs. control) significant differences were found in perceived dyspnea (p = 0.041), and right and left quadriceps muscle strength (p = 0.008 and p = 0.010, respectively). In addition, the subscale of “domestic activities” of the functional ability related to respiratory symptoms questionnaire showed significant differences (p = 0.036). EFFECTIVE |
López-López et al. 2019 [35] |
IG:48 (41.67) CG:47 (57.45) |
NR |
Fatigue (32.04 (18.58) vs. 46.22 (8.90)) and cough (18.84 (2.47) vs. 17.40 (3.67)) showed higher improvement in the intervention group, and significant differences were observed between the groups. An integrated programme of physical and electrical therapy during hospitalization improves physical and functional performance in patients with pneumonia. EFFECTIVE |
Ryrsø et al. 2024 [36] |
IG1:61 (49) IG2:63 (37) CG:62 (48) |
CURB-65, n(%) IG1: 0-1: 35(57); 2: 21(34); 3-5: 5(8) IG2: 0-1: 36(57); 2: 22(35); 3-5: 5(8) CG: 0-1: 36(58); 2: 21(34); 3-5: 5(8) |
Supervised in-bed cycling and booklet exercise had no effect on LOS. Similarly, exercise training had no statistically significant effect on 90-day readmission risk. NOT EFFECTIVE |
Author Response
This study aimed to determine the effectiveness of physiotherapy in patients with uncomplicated hospital-acquired pneumonia through a systematic review and meta-analysis. The systematic review and meta-analysis methods were conducted appropriately and are evaluated to be very good research procedure.
Comment 1: The reviewer read the articles selected by the authors and extracted the conclusions into a table below.
Study |
Sample Size (% male) |
Pneumonia severity |
Results |
Graham et al. 1978 [28] |
IG: 27 (51.85) CG: 27 (48.14) |
NR |
Pneumonia does not resolve faster with chest physiotherapy and intermittent positive pressure breathing. NOT EFFECTIVE |
Tydeman et al.1989 [29] |
IG: 12 (NR) CG: 20 (NR) |
NR |
Physical therapy remains unproven for the treatment of previously fit patients with uncomplicated pneumonia. NOT EFFECTIVE |
Noll et al.1999 [30] |
IG:11 (27.27) CG:10 (30) |
NR |
Although the mean duration of leukocytosis, intravenous antibiotic treatment, and length of stay were shorter in the adjunctive osteopathic manipulative treatment group, these measures did not reach statistical significance. However, the mean duration of oral antibiotic use reached statistical significance at 3.1 days in the treatment group and 0.8 days in the control group. NOT EFFECTIVE |
Noll et al. 2000 [31] |
IG:28 (50) CG:30 (53.3) |
SAPS, mean±SD IG:9.4±4.0 CG:9.6±3.5 |
The osteopathic manipulative treatment group had a significantly shorter duration of intravenous antibiotic treatment and a shorter hospital stay. EFFECTIVE |
Mundy et al. 2003 [32] |
IG:227 (44) CG:231 (44) |
PSI, n(%) IG: Low: 129(57); Moderate: 71(31); Hight:27(12) CG: Low: 123(53); Moderate: 65(28), High:43(19) |
Early mobilization (EM) of hospitalized adults with community-acquired pneumonia (CAP) reduces hospital length of stay. EM was defined as sitting or walking out of bed for at least 20 minutes during the first 24 hours of hospitalization. Progressive mobilization occurred each subsequent day during hospitalization EM was effective. EFFECTIVE |
Carratala et al. 2012 [33] |
IG:200 (66) CG:201(64.2) |
PSI, mean±SD IG:100.5±32.5 CG:101.1±31.5 |
3-step critical pathway including early mobilization and use of objective criteria for switching to oral antibiotic therapy and for deciding on hospital discharge or usual care was safe and effective in reducing the duration of intravenous antibiotic therapy and LOS for CAP and did not adversely affect patient outcomes. EFFECTIVE |
Martín-Sal-vador et al. 2016 [34] |
IG:24 (83.2) CG:20 (78) |
NR |
Between-groups analysis showed that after the physical therapy intervention (breathing exercises, electrostimulation, exercises with elastic bands and relaxation) (experimental vs. control) significant differences were found in perceived dyspnea (p = 0.041), and right and left quadriceps muscle strength (p = 0.008 and p = 0.010, respectively). In addition, the subscale of “domestic activities” of the functional ability related to respiratory symptoms questionnaire showed significant differences (p = 0.036). EFFECTIVE |
López-López et al. 2019 [35] |
IG:48 (41.67) CG:47 (57.45) |
NR |
Fatigue (32.04 (18.58) vs. 46.22 (8.90)) and cough (18.84 (2.47) vs. 17.40 (3.67)) showed higher improvement in the intervention group, and significant differences were observed between the groups. An integrated programme of physical and electrical therapy during hospitalization improves physical and functional performance in patients with pneumonia. EFFECTIVE |
Ryrsø et al. 2024 [36] |
IG1:61 (49) IG2:63 (37) CG:62 (48) |
CURB-65, n(%) IG1: 0-1: 35(57); 2: 21(34); 3-5: 5(8) IG2: 0-1: 36(57); 2: 22(35); 3-5: 5(8) CG: 0-1: 36(58); 2: 21(34); 3-5: 5(8) |
Supervised in-bed cycling and booklet exercise had no effect on LOS. Similarly, exercise training had no statistically significant effect on 90-day readmission risk. NOT EFFECTIVE |
Response 1: We acknowledge the reviewer comment and agree about the importance of presenting our data in the best way for readership. In this line we have included the suggested column to implement in table 2 as a conclusion.
Comment 2: The authors formulated the research question using the PICOS model.
There are five papers that do not specify the severity of pneumonia in the target population.
For physical therapy for pneumonia, the interventions that were ineffective were chest physiotherapy, physical therapy, adjunctive osteopathic manipulative treatment, supervised in-bed cycling, and booklet exercise.
On the other hand, the following interventions were effective: osteopathic manipulative treatment, early mobilization, physical therapy intervention (breathing exercises, electrostimulation, exercises with elastic bands, and relaxation), supervised in-bed cycling, and book exercise among the selected papers.
Response 2: We agree with the reviewer about the absence of pneumonia severity assessment in some of the studies. That fact can be attributed to the fact that some of those studies were published prior to the widespread adoption of PSI and CURB-65. Other manuscripts (published 2016 and 2019), focused on outcomes such as treatment response, rather than on severity stratification. We have now acknowledged this as a limitation in our review.
Comment 3: Although the selected studies were randomized clinical trials, two studies with very small sample sizes were included.
However, I believe that the conceptualization of physical therapy for uncomplicated hospital-acquired pneumonia, which is the focus of this study, has not been adequately established extracted papers result and conclusion.
Therefore, I believe that conclusions regarding the objectives of this study were not reached.
Response 3: We acknowledge the reviewer comment. In this line, we have included as a limitation that some of the studies in the review have small sample sizes, which may influence the robustness of the reported outcomes. Additionally, we understand that the extracted papers might not have fully captured the complexities of the intervention, and we agree that this could affect the clarity of our conclusions. Thus, we have included the results column from your table incorporated into Table 2 for clarification.
Comment 4: "What kind of intervention is physical therapy for pneumonia?"
By setting this research question, I believe this study will be able to produce optimal results.
Response 4: Thank you for your insightful comment. Physical therapy for pneumonia is considered a supportive and adjunctive intervention, aimed at improving pulmonary function, enhancing airway clearance, and preventing complications such as atelectasis and deconditioning. We appreciate your suggestion, and we agree that clearly the research question. Accordingly, we have clarified in the introduction section.
Comment 5: This study is an important study to clarify the clinical value and importance of physical therapy. I recommend conducting a thorough systematic review and working to establish a conceptual framework for physiotherapy in uncomplicated hospital-acquired pneumonia.
Response 5: As requested in Comment 4, we have clarified the conceptual framework of physical therapy for non-ICU pneumonia patients in the Introduction section.
Comment 6: Reference paper No. 34 is available in English. Please cite the English version as a reference.
Response 6: We have modified it as you suggested.
Reviewer 2 Report
Comments and Suggestions for Authors
The manuscript entitled “Effectiveness of physical y therapy interventions for non-ICU hospitalized pneumonia patients: A systematic review of randomized controlled trials” was interesting and well-written. The authors aimed to evaluate the efficacy of physical therapy in impacting health-related outcomes in hospitalized non-ICU pneumonia. However, some issues need further attentions.
1-In the methods section, the number and name of the databases are different from what has been said in the abstract.
2- In PICO criteria, population consists of adult patients. Please provide an age range and explain whether older adults were included or not.
3- Inclusion and exclusion criteria are not clear in the methods section.
4- The initial retrieved papers were 9460, and finally 9 articles were selected. This suggests that possibly search strategies were not specific enough. The search strategies in different databases should be uploaded as a supplementary file.
5- The discussion section needs to be expanded and include more studies to compare the results.
6- The conclusion section needs to be revised and expanded.
Author Response
The manuscript entitled “Effectiveness of physical y therapy interventions for non-ICU hospitalized pneumonia patients: A systematic review of randomized controlled trials” was interesting and well-written. The authors aimed to evaluate the efficacy of physical therapy in impacting health-related outcomes in hospitalized non-ICU pneumonia. However, some issues need further attentions.
Comment 1: In the methods section, the number and name of the databases are different from what has been said in the abstract.
Response 1: Thank you for your comment. We have corrected in the abstract.
Comment 2- In PICO criteria, population consists of adult patients. Please provide an age range and explain whether older adults were included or not.
Response 2: All patients aged 18 years and older were eligible for inclusion. We have clarified in the method section.
Comment 3: Inclusion and exclusion criteria are not clear in the methods section.
Response 3: We appreciate the reviewer comment. We have clarified the inclusion and exclusion criteria in the Methods section.
Comment 4: The initial retrieved papers were 9460, and finally 9 articles were selected. This suggests that possibly search strategies were not specific enough. The search strategies in different databases should be uploaded as a supplementary file.
Response 4: Thank you for your comment. As you suggested, we have included the search strategy in the supplementary file.
Comment 5: Thank you for your valuable comment. We have revised the discussion to incorporate additional relevant studies. These comparisons help to contextualize our findings and highlight similarities and differences with previous research.
Response 5: Thank you for your comment. As you suggested, we extended the discussion section.
Comment 6: The conclusion section needs to be revised and expanded.
Response 6: Thank you for your observation. In response to your suggestion, we have revised and expanded the conclusion section to better reflect the key findings of our study, emphasize the clinical implications, and highlight the need for future research.
Reviewer 3 Report
Comments and Suggestions for Authors
This manuscript demonstrated effectiveness of physical therapy interventions for non-ICU hospitalized pneumonia patients: a systematic review of randomized controlled trials.
However, major concern was founded in this manuscript. This is a lack of data extraction with 9 studies only. This is particularly problematic in systematic review.
Author Response
Comment 1: This manuscript demonstrated effectiveness of physical therapy interventions for non-ICU hospitalized pneumonia patients: a systematic review of randomized controlled trials. However, major concern was founded in this manuscript. This is a lack of data extraction with 9 studies only. This is particularly problematic in systematic review.
Response 1: Thank you for your thoughtful feedback. We acknowledge your concern regarding the limited number of studies included in our systematic review. As you correctly noted, a small number of studies may affect the comprehensiveness and generalizability of the findings. We have included as limitation. We have included our search strategy as a supplementary file and clarified the inclusion and exclusion criteria to ensure that all eligible studies were captured. While the number of included randomized controlled trials remains limited due to the scarcity of evidence specific to physiotherapy in non-ICU pneumonia patients. Besides, we have improved the main table of the results, and conclusions sections to more extended.